# Immune Checkpoint Inhibitors, Small-Molecule Immunotherapies and the Emerging Role of Neutrophil Extracellular Traps in Therapeutic Strategies for Head and Neck Cancer

**DOI:** 10.3390/ijms241411695

**Published:** 2023-07-20

**Authors:** Connor H. O’Meara, Zuhayr Jafri, Levon M. Khachigian

**Affiliations:** 1Department of Otorhinolaryngology, Head and Neck Surgery, Prince of Wales Hospital, Randwick, NSW 2031, Australia; 2Vascular Biology and Translational Research, School of Biomedical Sciences, UNSW Faculty of Medicine and Health, University of New South Wales, Sydney, NSW 2052, Australia

**Keywords:** immune checkpoint inhibitor, head and neck cancers, head and neck squamous cell carcinoma, neutrophil extracellular traps

## Abstract

Immune checkpoint inhibitor (ICI) therapy has revolutionized the treatment of many cancer types, including head and neck cancers (HNC). When checkpoint and partner proteins bind, these send an “off” signal to T cells, which prevents the immune system from destroying tumor cells. However, in HNC, and indeed many other cancers, more people do not respond and/or suffer from toxic effects than those who do respond. Hence, newer, more effective approaches are needed. The challenge to durable therapy lies in a deeper understanding of the complex interactions between immune cells, tumor cells and the tumor microenvironment. This will help develop therapies that promote lasting tumorlysis by overcoming T-cell exhaustion. Here we explore the strengths and limitations of current ICI therapy in head and neck squamous cell carcinoma (HNSCC). We also review emerging small-molecule immunotherapies and the growing promise of neutrophil extracellular traps in controlling tumor progression and metastasis.

## 1. Introduction

The immune system is a dynamic and equipped mechanism, an intricate system of “recognition” and “on-off” switches. Unfortunately, cancers utilize this system to enable growth and escape. The role of the immune system in tumor regulation is particularly evident in the immunocompromised. Iatrogenic solid organ transplant, diabetes, autoimmunity requiring immunosuppressive therapy, HIV and hemoproliferative malignant disease or disorders and aging, are all associated with an increased risk of developing head and neck cancer (HNC) and worse outcomes [1,2,3,4,5,6,7,8,9,10,11]. Proliferating tumors utilize many forms of immunosuppression to tip the balance of immunoediting toward tumor progression [12]. Identifying therapies capable of shifting this balance back toward immunosurveillance should play an integral role in reducing morbidity- and mortality-associated HNC.

HNC, the sixth most common group of malignancies worldwide, results in 680,000 new cases annually, with squamous cell carcinoma (SCC) being the most common [13,14,15]. The incidence of HNC is increasing due to a range of factors including smoking, alcohol, human papillomavirus (HPV) infection and extended life expectancy [16].

Despite the vital role played by traditional therapies for HNSCC, namely surgery, radiotherapy and chemotherapy, prognosis remains poor and survival remains correlated to stage, with a 5-year survival rate of 50–60% and more than 60% presenting in the advanced stage [17,18]. More than 50% of HNSCCs have tumor recurrence and metastasis in less than 3 years [19]. Targeting the epidermal growth factor receptor (EGFR) was hailed a paradigm shift in personalizing HNSCC treatment, with the monoclonal antibody cetuximab demonstrating promise [20,21]; however, this has since demonstrated limited efficacy [22].

Compared with traditional therapies, new immunotherapy agents, namely antibodies targeting the PD-1/PD-L1 system, so-called immune checkpoint inhibitors (ICI) provide improved efficacy and comparatively lower toxicity for patients with advanced HNSCC [23,24,25,26]. KEYNOTE-048 (NCT02358031), a randomized open-label phase 3 study comparing the humanized monoclonal antibody pembrolizumab (Keytruda) targeting PD-1 alone or in conjunction with chemotherapy (platinum and 5-fluorouracil) against cetuximab with chemotherapy, demonstrated overall survival improvement in both treatment arms over standard-of-care therapy in recurrent or metastatic HNSCC [27]. Pembrolizumab was subsequently approved as a first-line therapeutic drug for patients with metastatic, unresectable and recurrent HNSCC. Unfortunately, the objective response rate (ORR) of pembrolizumab (or nivolumab/Optivo) in HNSCC is only 15%, with short-term durability [28,29]. In addition, immune-related adverse events (irAEs) secondary to immunotherapy treatment were identified in over 50% of patients, impacting clinical outcomes [30], with adverse-event-associated mortality evident in 0.3–1.3% of patients [31]. Common irAEs include gastrointestinal, dermatologic and endocrine toxicities, more specifically dermatitis, rash, nausea/vomiting, fever, headache, myalgia, hypothyroidism and fatigue [32]. Rarely, irAEs can be severe, resulting in carditis, nephritis, hepatitis, pneumonitis, gastrointestinal perforation and severe hematological dysfunction [33]. irAEs in ICI therapy have been associated with benefits, namely improvements in PFS, OS and ORR [34,35,36,37]. Consequently, balancing immunotherapy de-escalation or commencement of immunosuppressive therapy against a sub-optimal oncological outcome can be difficult.

Predictive biomarkers may be the key to identifying patients at risk of irAEs. To date, circulating blood counts and ratios, autoantibodies and autoantigens, microRNAs, gastrointestinal microbiome, T-cell diversification and expansion and cytokines are all being investigated; however, they remain to be validated for clinical use [38].

Biological, etiological, phenotypic and clinical heterogeneities characterize HNSCC and challenge the development of personalized medicine. However, poor survival, significant morbidity and compromised quality of life emphasize the requirement for innovative therapy. Immunoediting is the process through which the immune system can promote and constrain tumor development [39]. This article explores current and developing therapies in immunomodulation and the developing role of neutrophil extracellular traps (NETs), net-like structures comprised of DNA-histone complexes and proteins in immune-mediated tumorigenesis.

## 2. Immune Checkpoint Inhibitor Targets and Therapies

A successful objective ICI response revitalizes the immune system to recognize and target cancer cells. The roles of known key immune checkpoints CTLA-4, PD-1 and LAG-3 are summarized in Figure 1.

### 2.1. CTLA-4 and PD-1/PD-L1

CTLA-4 (cytotoxic T-lymphocyte associated protein 4, also known as cluster of differentiation 152, CD152) and programmed cell death protein 1 (PD-1) (and its ligands PD-L1 and PD-L2) are immune checkpoints targeted by humanized antibodies for the treatment of HNSCC. CTLA-4 is bound by ipilimumab (Yervoy), whereas PD-1 is targeted by pembrolizumab and nivolumab [32]. The antibodies atezolizumab (Tecentriq), durvalumab (Imfinzi) and avelumab (Bavencio) have also been approved as inhibitors of PD-L1 [33]. Both checkpoints regulate different stages of the immune response. CTLA-4 is considered the “leader” of the immune response and prevents the stimulation of autoreactive T-cells in the initial stage of naïve T-cell activation, whereas PD-1 is thought to regulate previously activated T-cells at the later stages of the immune response [32].

CTLA-4 is a homolog of CD28, but unlike CD28, CTLA-4 activation has an immunosuppressive effect opposite to the stimulatory effect of CD28 and the T-cell receptor (TCR) [40]. The binding of CD80/CD86 on antigen-presenting cells to CTLA-4 on T-cells in the tumor microenvironment suppresses the immune system, enabling tumor proliferation [41]. PD-1’s interaction with PD-L1 and PD-L2 has an immunosuppressive effect [41]. PD-L1 and PD-L2 are expressed by a range of tumors including HNSCC [42]. Critically, increased PD-1 levels serve as a biomarker for T cell exhaustion; this state of exhaustion is linked to T-cell dysfunction, which can facilitate tumor proliferation [43]. PD-L1′s interaction with PD-1 has an immunosuppressive effect, thus protecting cancer cells from lysis by activated T-cells [44].

Despite ICI therapy demonstrating survival advantage, comparatively few patients develop an effective response, the durability of which attenuates with acquired tumor resistance. Acquired resistance leads to tumor progression, and both arms of the immune system, innate and adaptive, can play a critical role in this change. Mechanisms of resistance to immunotherapy can be either intrinsic (tumor cell-mediated) or extrinsic (processes associated with T-cell activation) and shift the balance of immunomodulation towards tumor proliferation. Intrinsic resistance can include the downregulation of antigen-presenting machinery (APM) [45], the up-regulation of signaling pathways promoting T-cell exhaustion [46], the expression of multiple checkpoint inhibitors to mitigate T-cell activation [47], changes in tumor cell DNA repair, damage and genomic instability [48] and altered kinase signaling pathways [49]. Extrinsic resistance involves the complex interplay between tumor cells and the tumor microenvironment and its ability to regulate phenotypical characteristics of immune cells, especially TANs, TAMs, Tregs, MDSCs, T-cells, their associated regulatory cytokines and signaling pathways and a newly identified player, NETs [50,51,52,53,54,55,56].

Despite the clear improvements in overall survival due to immune checkpoint therapy, such treatments have limitations. For example, since CTLA-4 prevents the stimulation of autoreactive T-cells, inhibiting CTLA-4 can lead to grade 3 or 4 autoimmune-related adverse effects in 10–15% of patients [57]. Immune checkpoint immunotherapies are also associated with low response rates. For example, pembrolizumab has a response rate of only 15% in HNSCC [58].

To improve therapeutic failure and overcome immunotherapy resistance, significant energy is being invested in exploring biomarkers to predict clinical response and combinational therapies or changes in adjuvant delivery of immunotherapy to increase success rates. Biomarkers that have shown potential to determine improved clinical response in HNSCC include the tumor mutational burden, CCND1 amplification (CCND1 encodes cyclin D1, which regulates the retinoblastoma protein activity and cell-cycle progression), PD-1, IFN-γ, tumor-infiltrating lymphocytes (TILs) and cancer-associated fibroblasts (CAFs), CTLA-4, exosomes, CXCL, MTAP and SFR4/CPXM1/COL5A1 molecules [25,59,60,61,62,63,64,65,66,67,68,69].

Clinical trials exploring combinational immunotherapy in HNSCC are underway. The phase 3 randomized trial CheckMate 651 NCT02741570), which compared nivolumab and ipilimumab against EXTREME (platinum/5-fluorouracil/cetuximab) for R/M HNSCC, was unsuccessful in demonstrating OS improvement, although there was an association between elevated CPS and OS and durable response [70]. Other combination ICI therapy clinical trials have been largely unsuccessful (Table 1).

Concurrent neoadjuvant and adjuvant delivery of ICIs has recently demonstrated benefits in surgically resectable advanced melanoma (Stage IIIB to IVC). In a recently completed Phase 2 randomized study (NCT03698019), neoadjuvant-adjuvant delivery of pembrolizumab was compared to an adjuvant alone in demonstrating an event-free survival of 72% in the neoadjuvant-adjuvant group compared to 49% in the adjuvant group after 2 years [76].

### 2.2. LAG-3

LAG-3 is expressed on activated human T-cells and natural killer cells and plays a similar role in T-cell regulation to CTLA-4 and PD-1 [77]. LAG-3 may represent an intrinsic resistance mechanism to PD-1 inhibitors due to its synergistic co-expression with PD-1 on exhausted T-cells [77]. To combat resistance, the FDA-approved drug opdualag^®^ (combined LAG-3 and PD-1 inhibitor) became a first-line treatment for unresectable or metastatic melanoma in March 2022 [77]. Opdualag has shown success in clinical trials, more than doubling progression-free survival compared to melanoma patients treated with nivolumab alone [78].

### 2.3. Tim-3 and CD39

T cell immunoglobulin and mucin domain-containing protein 3 (Tim-3) is a co-inhibitory receptor expressed on IFN-γ-producing T-cells Tim-3. Studies by Liu et al. showed that Tim-3 is linked to immunosuppression in HNSCC and that targeting Tim-3 (with monoclonal antibodies) can enhance the anti-tumor immune response by reducing Tregs in HNSCC [79]. Similarly, the expression of the cell-surface ectonucleosidase CD39 in HNSCC positively correlates with tumor stage and predicts poor prognosis [80]. There are no approved inhibitors of Tim-3 or CD39, and opdualag has not yet been approved for HNSCC.

## 3. Emerging Immunotherapeutic Targets and Strategies

Small-molecule immunotherapy (SMI) may represent the paradigm shift required to improve quality of life (QOL) and survival in HNSCC. Unlike current ICI therapies, small molecules can be delivered orally, are potentially less expensive than antibodies and utilized to target intracellular signaling and transcriptional pathways upstream of receptors expressed on the cell surface. Several promising SMIs in various phases of development are listed in Table 2 and described below.

### 3.1. STAT3 

Among the intracellular signal transducer and activator of transcription (STAT) proteins, STAT3 plays an important hemostatic role in normal cells by helping to regulate cell growth, survival, differentiation, angiogenesis, immune response and cellular respiration [87,88]. STAT3 can be activated by both JAK and EGFR (via Src) signaling, subsequently binding target DNA to regulate gene expression [89,90]. Importantly, STAT3 activation can upregulate multiple survival proteins, namely Bcl-xL, survivin and Bcl-2 [91]. STAT3 is also an upstream regulator of PD-1 expression [92,93]. STAT3 can behave as an oncogene and is expressed in approximately 70% of human cancers [19]. Overexpression of STAT3 regulates tumor progression and is associated with poor prognosis in various malignancies [94]. Elevated levels of IL-6, released by tumor-infiltrating lymphocytes, M2 phenotype macrophages [95], cancer-associated fibroblasts [96] and tumor cells [97,98], can activate a pro-tumorigenic IL-6/STAT3 pathway [99], inhibiting dendritic cell maturation, suppressing CD8^+^ T-cell and NK cell activation [100,101,102] and promoting CD4^+^ T cell differentiation to a T regulatory phenotype [103]. This pathway supports the survival of immunosuppressive MDSCs and M2 phenotype macrophages [104,105] and supports tumor survival, invasiveness and proliferation [106]. STAT3 can also regulate metabolism-related genes that favor cancer progression [107] and promote angiogenesis via the upregulation of VEGF [108]. STAT3 activation was identified to be 10.6- and 8.8-fold higher in tumors and normal mucosa, respectively, of HNSCC patients compared to the mucosa of non-cancer patients, supporting the concept of “field cancerization” [109]. Furthermore, there is a strong association between downstream proteins transcribed by STAT3 and locoregional metastasis, stage, recurrence and mortality in OCSCC [110]. There is also evidence that constitutive STAT3 activation plays a prominent role in mediating drug resistance to many targeted cancer therapies and chemotherapies, including the poor response to the EGFR blockade, with less than 15% of patients benefiting from cetuximab as a single agent or 36% when combined with chemotherapy [111]. Certainly, STAT3 inhibition has improved radiotherapy and cetuximab responsiveness in HNSCC cell lines [91,112]. Hence preclinical and early clinical trial data suggest that targeting STAT3 is a promising therapeutic strategy [113].

Flubendazole is a benzimidazole, long utilized as a macrofilaricide in humans and animals. Recently, it has been repurposed and recognized as a promising anti-cancer agent, effective in breast cancer, melanoma, neuroblastoma, colorectal, liver and squamous cell carcinoma [114,115,116,117,118,119,120,121,122,123,124]. In melanoma, flubendazole reduced the expression of phosphorylated STAT3 in tumor tissue and the expression of PD-1 expression, while also decreasing MDSC levels in tumors [125]. Immunological signature gene sets, including those associated with T cell differentiation, proliferation and function correlated with FLU treatment [126]. Flubendazole has also been found to have synergistic antiproliferative effects in vitro with 5-fluorouracil [127], which raised the potential benefit of use topically in conjunction with 5-FU for the treatment of premalignant and malignant non-melanoma skin cancers [128,129,130]. Flubendazole has not entered clinical trials for HNC.

An alternative approach is provided by danvatirsen (AZD9150), an antisense oligonucleotide inhibitor of STAT3 comprised of 16 nucleotides, which has demonstrated antiproliferative effects in xenograft models showing reduced STAT3 expression, paving its way to clinical trial [131]. Combining durvalumab (MEDI4736, PD-L1 inhibitor) with danvatirsen or AZD5069 (CXCR2 inhibitor) (NCT02499328) in patients with advanced solid malignancies and HNSCC improved anticancer activity as compared to PD-L1 monotherapy [81].

### 3.2. STING

STING (stimulator of interferon genes) is a cytosolic pattern-recognition receptor (PRR) that recognizes non-self-dsDNA, upregulating type 1 interferon [132]. Recent evidence indicates that type 1 IFN plays an important role in many anticancer modalities, including immunotherapy, helping to promote dendritic cell activation and prime and recruit cytotoxic CD8^+^ T-cells against tumor-associated antigens [133,134,135]. Evidence suggests that the STING pathway may help potentiate checkpoint blockade therapy [136,137].

### 3.3. PPAR 

Peroxisome proliferator-activated receptors (PPARs) regulate a multitude of cellular functions. They are a family of ligand-inducible nuclear hormone receptors belonging to the steroid receptor superfamily. PPAR-α is commonly expressed in skeletal muscle, liver, heart and brown adipose tissue. Its activation suppresses NF-κB signaling, which decreases the inflammatory cytokine production by different cells and modulates the proliferation, differentiation and survival of macrophages, B-cells and T-cells, whilst also playing a role in angiogenesis, homeostasis and glucose and lipid metabolism [138,139,140,141]. Notwithstanding a pleiotropic role in cancer, which appears type and tumor microenvironment (TME)-dependent, increasing evidence is demonstrating that PPAR-γ can modulate carcinogenesis, showing promise as a focus for cancer therapies. PPAR-γ agonists have been shown to inhibit cancer cell proliferation and Warburg effects.

### 3.4. RTKs

There are over 50 known RTKs in humans. These are transmembrane receptors integral to cell-to-cell communication and the regulation of cell growth, metabolism, motility and cell differentiation. These mediate the activation of a variety of signaling pathways, including JAK/STAT, PI-3K/AKT/mTOR, PLC/PKC and RAS/MAPK. Their dysregulation plays a role in multiple human disease processes, including carcinogenesis, which can confer constitutive activation by genomic amplification, chromosomal rearrangements, autocrine activation, gain-of-function mutations or kinase domain duplication [142,143].

There is evidence that IL-33, although a pleiotropic cytokine in HNSCC [144], can regulate immune cells in the TME, namely CD4^+^ T-helper cells, CD8^+^ T-cells, NK cells, DCs and macrophages [144]. Developing evidence suggests that IL-33 may regulate the immune response through a signaling complex between IL-33R and EGFR in gastrointestinal helminth infections [145]; however, we are not aware of current research supporting this pathway in cancer.

EGFR is a prototypic RTK and is well recognized to be susceptible to gain-of-function mutations and is commonly overexpressed in HNSCC. These mutations can hyperactivate the kinase and its downstream signaling, conferring oncogenic properties [146]. Eighty to ninety percent of HNSCCs overexpress or demonstrate EGFR mutation, with these changes detrimentally affecting both PFS and OS [147,148]. Certainly, EGFR status has been identified as a survival predictor and guide to the effectiveness of chemoradiation [149]. Although known mutations in EGFR are rare in HNC, its overexpression with TGF-α is common, and auto or paracrine activation is important in HNC EGFR function. Unfortunately, mutation commonly alters drug binding dynamics, leading to resistance, a similar phenomenon leading to a reduced radiation response and overall survival in HNSCC. Mutation status is also associated with the tumor stage [150]. Chromosomal rearrangements have been identified in the RET kinase in thyroid cancer [151] and TRKA, TRKB and TRKC tyrosine kinases in thyroid and HNC.

The primary site of action of TKIs on EGFR is the intracellular tyrosine kinase domain, inhibiting downstream signaling. Geftinib and erlotinib have been ineffective in HNC, comparatively lapatinib, afatinib and dacomitinib have demonstrated benefits in clinical trials and can target VEGFR to reduce tumor angiogenesis. Cetuximab has shown an ORR of 13% as a monotherapy, and gefitinib and erlotinib demonstrated ORR of 1.4% and 10.6% with a median OS of 5.5 and 8.1 months, respectively [152,153,154]. Anlotinib (AL3818) is a novel multi-target RTK antagonist against PDGFR, FGFR, VEGFR and c-Kit. In human OCSCC cell lines, it effectively reduced tumor cell proliferation and promoted apoptosis [155].

### 3.5. AHR 

AHR (aryl hydrocarbon receptor) is a ligand-activated transcription factor activated by both anthropogenic and natural agonists, with recent studies reporting a key role in regulating host immunity [156,157]. As a transcription factor, AHR can regulate the expression of cytochrome P450 family genes. Chronically active AHR is capable of driving cancer cell invasion, migration, cancer stem cell characteristics and survival [158]. Tumor expression of AHR can result in an autocrine AHR-IL-6/STAT3 signaling loop via kynurenine, an immunosuppressive AHR agonist ligand produced by the metabolism of the essential amino acid tryptophan [159].

## 4. Neutrophils Extracellular Traps (NETs)

Neutrophils are the largest group of leukocytes within the blood and play an integral role in immune-mediated host defense mechanisms. As activated phagocytes, these secrete neutrophil elastase (NE), reactive oxygen species (ROS), nicotinamide adenine dinucleotide phosphate oxidase (NADPH) and myeloperoxidase (MPO) to digest pathogens [160]. Chemotactically attracted to the TME, tumor-associated neutrophils (TANs) phenotypically polarize to either N1 or N2 sub-types. Similar to M1 phenotype tumor-associated macrophages (TAMs), N1 TANs are anti-tumorigenic, whilst N2 TANs are pro-tumorigenic and regulate immunosuppression, tumor cell proliferation, angiogenesis and metastasis [161,162,163].

Derived from neutrophils undergoing a signal-mediated cell death program known as NETosis, NETs are extracellular “spider webs” of unwound chromatin, comprising histones, neutrophil elastase and granular antimicrobial enzymes [164,165] (Figure 2). These are key antimicrobial components of the innate defense system that sequester and contribute to bacterial cytotoxicity and phagocytosis [166,167]. Their antimicrobial purview includes inhibiting replication, containing and eliminating viral infections via the activation of TLR4, 7 and 8 pathways, PKC pathway blockade or aggregation and neutralizing effects of cationic histones, particularly arginine-rich H3 and H4 [168,169,170,171,172]. NETs are important in clearing large pathogens and are activated by β-glucan on fungal hyphae [173,174,175]. NETs recognize the activation of platelets and monocytes and limit the dissemination of parasites by trapping and killing these with histones, neutrophil elastase, MPO and collagenase mediating cytotoxicity [176,177,178,179]. Unfortunately, NET dysregulation is pivotal to the pathogenesis of numerous diseases, including sepsis [180,181], acute respiratory disease syndromes [182,183], ischemia-reperfusion injury [184], diabetes [185], venous thromboembolism [186] and chemotherapy-induced peripheral neuropathy [187].

New evidence indicates that NETs potentiate pro-tumorigenic effects, with neutrophils attracted to the tumor microenvironment being reprogrammed by tumor-associated factors to undergo NETosis and potentiate tumor activity. In this regard, NETs are capable of suppressing tumor cell apoptosis and promoting tumor cell invasion [188,189,190,191,192,193,194,195]. Factors involved in tumor-mediated NET formation include tumor-derived inflammatory and chemoattractant cytokines (IL-8, IL-6, TNF-α, G-CSF and IL-1β) [189,190,196,197,198,199], tumor extracellular vesicles [200], tumor-activated platelets [201,202], tumor-derived HMGB1 [203,204,205,206], KRAS oncogene mutation [207] and hypoxia [208].

Recent evidence has highlighted the ability of NETs to actively drive tumor growth and metastasis. NET-associated HMGB1 promotes tumor cell proliferation involving interaction with tumor RAGE, activating and NF-κB signaling [209]. Additionally, neutrophil elastase, via the PI3K signaling pathway, promotes the proliferation of adenocarcinoma cells [210]. NETs play a key role in shielding tumor cells from tumor-recognizing NK- and cytotoxic CD8^+^ T-cells [211,212], whilst promoting cytotoxic CD8^+^ T-cell exhaustion through the upregulation of PD-L1 [213]. Their immunosuppressive role also extends to the programming of T regulatory phenotype cells, which inhibit macrophage, dendritic, cytotoxic CD8^+^ T cell anti-tumor effects via a TLR4-dependent mechanism [214,215,216], which may be histone dependent, while they also play a key role in thrombosis [217,218,219,220,221]. Further compromising immunorecognition, NET-activated platelets may facilitate plasma membrane transfer, enabling tumor cell expression of platelet markers and MHC receptors to camouflage their presence within the platelet aggregate attached to a NET scaffold [222,223,224].

NETs can affect all arms of Virchow’s triad [225], activating platelets and endothelial cells, aggregating erythrocytes [226] and promoting tissue factor release from both platelets and endothelial cells. Specifically, by way of histone expression, NETs induced an endothelial cell shift toward a pro-coagulant phenotype [227,228,229]. NETs also promote thrombosis by acting as a scaffold that can capture and activate platelets, facilitate fibrin deposition and express TF for coagulation [230]. There is likely significant crosstalk between NETs and platelets, working in concert to enhance tumor cell survival and metastasis. Platelet TLR4 can trigger NET production, and NET-expressed histone H3 and H4 can activate platelets in a positive feedback loop [217,231].

The ability of NETs to promote thrombosis and capture circulating tumor cells enhances the ability of tumor cells to metastasize. However, their further ability to facilitate distant tumor growth is much more extensive. NETs can promote tumor cell migration and enhance invasiveness, with the rearrangement of the cytoskeleton elements, via their CCDC25 receptor, which may aid tumor cell transmigration across the endothelium [232]. Although the mechanism remains unclear, NETs can reprogram the epithelial-to-mesenchymal transition in tumor cells, essentially allowing them to disconnect cell-to-cell and cell-to-extracellular matrix chelation and commence migration and invasion [233,234,235]. NETs can also revive dormant tumor cells that have metastasized to an unfavorable microenvironment. Evidence suggests that extracellular matrix NET remodeled laminin binds dormant tumor a3B1 integrin, driving their activation via FAK/ERK/MLCK/YAP signalling, in turn supporting proliferation [236].

Angiogenesis is also a priority for hypoxic tumor cells. NETs have been demonstrated to promote angiogenesis [237,238,239] and this may well be via a histone-dependent mechanism [237]. Aldabbous and colleagues showed that NETs increased the vascularization of Matrigel plugs and release of MMP-9, TGF-β1 latency-associated peptide, HB-EGFGF and uPA, also promoting endothelial permeability and cell motility [238].

Interactions between NETs and cancer cells are also thought to drive resistance to various cancer therapies, including chemotherapy, immunotherapy and radiation therapies. Therefore, the development of therapies to mitigate the pro-tumorigenic role of NETs is an absolute necessity but should minimize interference with immunity, wound healing, and host defense mechanisms. There are several novel therapies being developed for the management of NETs in sterile systemic inflammatory response syndromes and several agents that may be repurposed to mitigate the pro-tumorigenic role of NETs [240,241,242,243].

## 5. Potential NET-Based Therapies

### 5.1. Novel Compounds Facilitating NET Prevention or Modulation

#### 5.1.1. Conceptual

STC3141 (methyl β-cellobioside per-O-sulfate) is a small polyanion (SPA) that interacts electrostatically with histones, neutralizing their pathological effects preclinically in several pathologies [240]. This agent was developed specifically to target histones on NETs, to help preserve the host defense benefit of the protease-labelled chromatin web and facilitate microbe cytotoxicity in sepsis (histone neutralization with NET stabilization). Phase 1b results demonstrate favorable safety profiles and clinical benefits in ARDS. STC3141 is likely to interfere with the pro-tumorigenic functions of NETs, including tumor cell camouflage, migration and dormant cell reactivation, and may represent an effective small-molecule NET modulator. STC3141 may inhibit histone-dependent pathways, including TLR4/histone-dependent TME immunosuppression, histone-dependent endothelial and platelet activation and thrombosis, conferring survival and metastatic ability.

#### 5.1.2. Preclinical

Sivelestat is an inhibitor of the NET-expressed serine protease neutrophil elastase, competitively inhibiting it with high specificity. NE plays a key role in NETosis and NET formation, and the pro-tumorigenic role of NE has been confirmed in breast, lung and colon cancers [244,245,246]. In keeping with the role of NETs in metastasis, Okamoto and colleagues demonstrated that sivelestat reduced NET formation and liver metastasis in a murine model of colorectal cancer (CRC) but had no effect on primary tumor growth or the suppression of liver metastasis if the CRC cells had already metastasized [247].

GSK383 chloramidine is a PAD4 inhibitor. PAD4 is a peptidyl arginine deiminase type IV enzyme, critical to the formation of NETs [248]. In 4T1 murine breast cancer cells, co-culture with GSK383 significantly attenuated NET production and inhibited NET-mediated tumor cell invasion.

#### 5.1.3. Clinical Trials

Given the importance of the CXCR1/2:IL-8 axis in neutrophil/NET-mediated carcinogenesis, there are currently several CXCR1/2 inhibitors undergoing clinical trials in combination with ICIs. SX-682 (a CXCR1/2 antagonist) in combination with pembrolizumab entered Phase 1 trials in metastatic melanoma (NCT03161431), while a combination of avarixin (CXCR1/2 antagonist) and pembrolizumab is being trialed in advanced/metastatic solid tumors in a Phase 2 study (NCT03473925). CXCR1/2 inhibitors are not currently being trialed in HNC.

### 5.2. Repurposed Compounds Facilitating NET Prevention or Modulation

#### Preclinical/Clinical Cohort Studies

Aspirin is a COX-1 inhibitor commonly utilized as an antagonist to the primary hemostatic role of platelets. In a murine model of lipopolysaccharide-induced lung injury, aspirin reduced target tissue invasion by neutrophils and NET production. This is thought to be mediated by the amelioration of platelet-dependent release of CXCL4 (PF4) and CCL5 (RANTES), both of which increase neutrophil recruitment. Low-dose aspirin can have an anti-metastatic effect via a COX-1 inhibition-mediated reduction in NET production [249,250,251].

Metformin is a PKC inhibitor that attenuates NETosis. Previous studies identified that circulating NE, citrillinated histone, ds-DNA and proteinase-3 levels were reduced in the presence of metformin, and furthermore, metformin decreased expected NETosis in the presence of stimuli [252]. New evidence in hepatocellular carcinoma (HCC) and pancreatic cancer (PC) demonstrated that metformin attenuated NET production and reduced the metastatic potential of HCC and PC cells [253,254]. The role of metformin in attenuating NET-mediated carcinogenesis was corroborated in further murine models [255]. Interestingly, a recent presentation at AACR’s Annual Meeting revealed that patients with type 2 diabetes mellitus suffering from colorectal cancer had significantly improved DFS in the presence of metformin, with tissue analysis identifying a significant reduction in tumor-associated NETs and a significant increase in CD8^+^ T-cells. The authors concluded that metformin inhibits neutrophil infiltration and NET expression whilst promoting the infiltration of cytotoxic CD8^+^ T-cells in the TME [256].

### 5.3. Compounds Facilitating NET Destruction

#### Preclinical

Unfractionated heparin (UH) is a glycosaminoglycan that potentiates the enzyme antithrombin III, inactivating thrombin, factor Xa and other proteases. It has a high affinity for extracellular histones and has been shown to promote the degradation of NETs [257]. UH is readily available and no longer under patent but has an off-target side effect in increasing the risk of bleeding.

Dornase alfa (rhDNase 1) is a recombinant human deoxyribonuclease, which selectively cleaves DNA. A co-culture of triple-negative breast cancer cells with neutrophils formed significant NETs, with DNase 1 blocking both NET-mediated cancer cell migration and invasion. In vivo, DNase 1 has demonstrated some ability to attenuate metastasis in a murine lung cancer model [258]. Park and colleagues concluded that this effect could be improved by increasing the half-life of DNase 1 and developing DNase-1-coated nanoparticles, which reduced the metastatic burden in a 4T1 breast cancer murine model [259]. Wang and colleagues also demonstrated that rhDNase 1 mitigated the pro-tumorigenic effects of NETs in murine pancreatic cancer [253]. rhDNase 1 is an FDA-approved therapy to degrade chromatin in cystic fibrosis and is currently undergoing a clinical trial to determine similar benefits in COVID-19-associated ARDs (NCT04402944) [260].

## 6. Conclusions

While tumor development may be controlled by cytotoxic adaptive and innate immune cells, the challenge to durable therapy lies in understanding the complex interaction between the immune cells, the tumor and its microenvironment and delivering therapies that re-program immunomodulation toward tumorlysis. In this article, we explored the strengths and limitations of current ICI therapies and outlined emerging SMI targets and modalities including our growing understanding of the important role NETs play in immunomodulation and their promise in ameliorating tumor progression and metastasis.

## Figures and Tables

**Figure 1 ijms-24-11695-f001:**
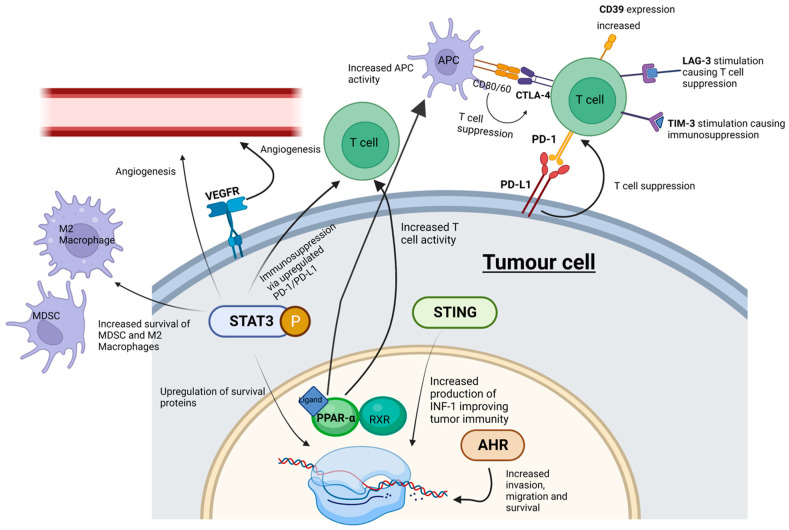
ICI and SMI actions within the tumor microenvironment. Whilst ICIs influence cell signaling at cell surface receptors, SMIs can interact with “upstream” intracellular signaling pathways-potentially playing a more effective role in abrogating tumor cell progression. MDSC, myeloid-derived suppressor cells; M2 macrophages, pro-tumorigenic macrophages; STING, stimulator of interferon genes; PPAR-α, peroxisome proliferator-activated receptor-α; AHR, aryl hydrocarbon receptor; STAT3, signal transducer and activator of transcription 3; P, phosphorylation of STAT3. Created with BioRender.com.

**Figure 2 ijms-24-11695-f002:**
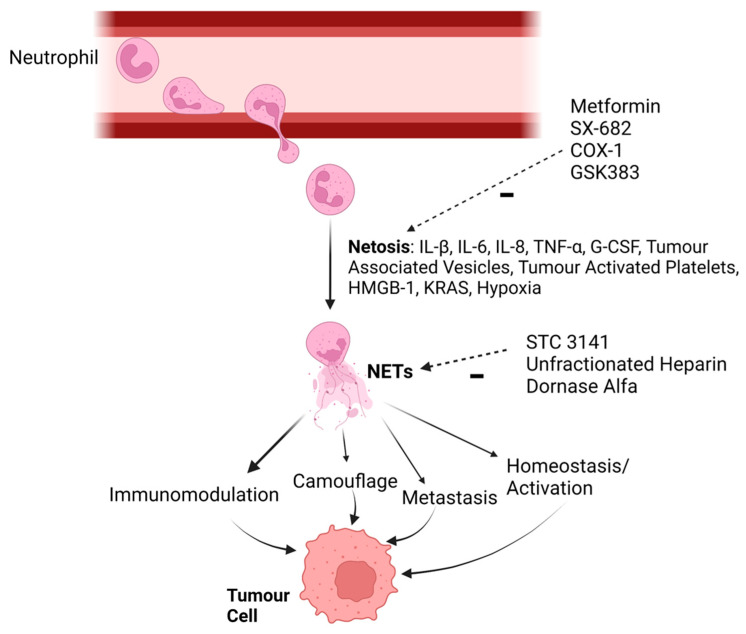
Pro-tumorigenic effects of NETs and developing therapeutic strategies. NETs have been demonstrated to play a role in tumor immunomodulation, camouflage, homeostasis/activation, and metastasis. Therapies under development can be grouped into those that prevent or “modulate” NETosis (the generation of NETs) or “dissolve” the chromatin “web” of the neutrophil extracellular trap, mitigating the ability for NETs to play a pro-tumorigenic role. HMGB-1, high mobility group box-1; KRAS oncogene mutation. Created with BioRender.com.

**Table 1 ijms-24-11695-t001:** Combination ICI Therapy Clinical Trials in HNSCC.

Target	Combination	Phase	Trial	Intent	Outcome
PD-1, CTLA-4	Nivolumab, Ipilimumab	3	NCT027441570 (CheckMate 651) [71]	Combination nivolumab + ipilimumab vs. EXTREME Regime (platinum/5-fluorouracil/cetuximab) for R/M HNSCC	Failed endpoint (OS). No difference between dual ICI blockade and EXTREME arm. Improvement in dual ICI arm if CPS > 20 (ns)
PD-L1, CTLA-4	Durvalumab, Tremelimumab	3	NCT02551159 (KESTRAL) [72]	Combination durvalumab + tremelimumab vs. duravalumab monotherapy vs. SOC CT in R/M HNSCC	Results pending
PD-1, CTLA-4	Nivolumab, Ipilimumab	2	NCT02823574 (CheckMate 714) [73]	Combination nivolumab + ipilimumab vs. nivolumab + ipilimumab placebo in R/M HNSCC	Failed ORR and OS endpoints. Subpopulation assessment ongoing.
PD-L1, CTLA-4	Durvalumab, Tremelimumab	3	NCT02369874 (EAGLE) [74,75]	Combination durvalumab + tremelimumab vs. durvalumab monotherapy vs. SOC in R/M HNSCC	Failed to meet primary OS improvement endpoint

**Table 2 ijms-24-11695-t002:** SMI Targets and Clinical Trials in HNSCC.

Target	Drug	Phase	Trial	Intent	Outcome
STAT3	AZD9150	1b/2	NCT02499328 [81]	Combination ASD9150 + MED14736 (duravalumab) vs. MED14736 alone; in platinum refractory recurrent metastatic HNSCC	Acceptable toxicity profile. Combination therapy more effective than PD-L1 monotherapy
STING	MK-1454	1	NCT03010176 [82]	Combination MK-1454 (ulveostinag) + pembrilizumab vs. MK-1454 monotherapy; in advanced HNSCC	Acceptable toxicity profile. Combination therapy more effective (DCR 48%) than monotherapy (DCR 20%)
PPAR-α	TPST-1120	1/1b	03829436 [83]	Combination TPST-1120 + nivolumab vs. TPST-1120 monotherapy; in advanced solid tumors; including HNSCC	Acceptable toxicity (several patients suffered Grade 3 Adverse reactions. Optimal disease response in combination therapy (38%)
RTKs	AL3818	2	NCT04999800 [84]	Combination AL3818 (analotinib) + pembrolizumab; as a first line therapy for platinum refractory recurrent or metastatic HNSCC	Manageable side effects. Encouraging anti-tumor activity. ORR: 46.7% (7/15) & DCR: 100% Median PFS & OS not reached (median follow-up: 8.2 months.
RTKs	Afatinib	3	NCT01345682 [85]	LUX-Head & Neck 1: second-line afatinib therapy vs. methotrexate for platinum refractory recurrent/metastatic HNSCC	*n* = 483 patients. Median PFS: afatinib over methotrexate (2.7 months vs. 1.6 months) Afatinib more effective in all tumor subsets except HPV + OPSCC
AHR	BAY2416964	1	NCT04069026 [86]	AHR antagonist: safety and tumor response study in advanced HNSCC & nSCLC	Well tolerated at all dose regimes. Initial evaluation of biomarkers shows inhibition of AHR and modulation of immune functions. Encouraging preliminary anti-tumor activity in heavily pretreated patients.

## Data Availability

Not applicable.

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
