# Peer review of "Immune Checkpoint Inhibitors, Small-Molecule Immunotherapies and the Emerging Role of Neutrophil Extracellular Traps in Therapeutic Strategies for Head and Neck Cancer"

_ijms, 2023, doi:10.3390/ijms241411695_

Round 1

Reviewer 1 Report

  1. Lack of clarity in sentence structure: Some sentences in the review are long and convoluted, making it difficult to understand the intended meaning. The author should consider breaking down complex sentences into smaller, more concise statements.

  2. Inconsistent capitalization and formatting: The capitalization of certain terms and names is inconsistent throughout the review. It is important to adhere to a consistent formatting style for clarity and professionalism.

  3. Missing information: The review mentions "outcomes from the KEYNOTE-048 trial" without providing any details about the trial or its findings. It is important to provide relevant information about studies or trials mentioned to give readers a complete understanding.

  4. Incomplete information: The review briefly mentions adverse events associated with immunotherapy treatment but fails to provide specific examples or details. Providing more information about the types of adverse events and their impact on patients would enhance the review's credibility.

  5. Lack of discussion on limitations: The review discusses the limitations of immune checkpoint immunotherapies in terms of autoimmune-related adverse effects and low response rates. However, it does not delve into potential strategies or ongoing research to overcome these limitations. Providing a more comprehensive discussion on the limitations and future directions would strengthen the review.

  6.  

Quality of English is good.

Author Response

Reviewer 1

  1. Lack of clarity in sentence structure: Some sentences in the review are long and convoluted, making it difficult to understand the intended meaning. The author should consider breaking down complex sentences into smaller, more concise statements.

We thank the reviewer for their helpful suggestion.  We have shortened sentences for greater clarity throughout the revised manuscript. 

  1. Inconsistent capitalization and formatting: The capitalization of certain terms and names is inconsistent throughout the review. It is important to adhere to a consistent formatting style for clarity and professionalism

We thank the reviewer for their helpful suggestion.  We have made capitalization and formatting consistent throughout the revised manuscript. 

  1. Missing information: The review mentions "outcomes from the KEYNOTE-048 trial" without providing any details about the trial or its findings. It is important to provide relevant information about studies or trials mentioned to give readers a complete understanding.

We thank the reviewer for their helpful suggestion.  We have now provided detail in regard to outcomes from the KEYNOTE-048 trial (lines 64-67).

  1. Incomplete information: The review briefly mentions adverse events associated with immunotherapy treatment but fails to provide specific examples or details. Providing more information about the types of adverse events and their impact on patients would enhance the review's credibility.

We thank the reviewer for their helpful suggestion.  We have now provided more information in regard to immune-related adverse events (irAEs) secondary to immunotherapy (lines 71-84).

  1. Lack of discussion on limitations: The review discusses the limitations of immune checkpoint immunotherapies in terms of autoimmune-related adverse effects and low response rates. However, it does not delve into potential strategies or ongoing research to overcome these limitations. Providing a more comprehensive discussion on the limitations and future directions would strengthen the review.

We thank the reviewer for their helpful suggestion.  We have now provided more information in regard to potential strategies/research to overcome low response rates and combination clinical trial information (lines 211-237).

Reviewer 2 Report

Immune Checkpoint Inhibitors, Small Molecule Immunotherapies and the Emerging Role of Neutrophil Extracellular Traps 3 in Therapeutic Strategies for Head and Neck Cancer

Comment

1. Line -13 Immune checkpoint inhibit (ICI) in line 47 is Immune checkpoint inhibitor (ICI) abbreviation check it.

2. Line 75, CTLA-4 (cytotoxic T-lymphocyte associated protein 4) the other name called CD152 needs to be change the sentence.

3. Line 83 the word “naïve” have the vowel “i” any special meaning?

4. The missing part of this review is a table. For better understanding, for reader there must be one table. For reference, below review article for references and it will help you to make a table.

·       10.3390/ijms241310546,

·       10.1038/s41568-019-0224-7,

·       10.3389/fimmu.2021.676301

5. If the clinical trial has a better explanation with all details it will be better follow the articles links in comment 4 and try to elaborate. 

Good.

Author Response

Reviewer 2

  1. Line -13 Immune checkpoint inhibit (ICI) in line 47 is Immune checkpoint inhibitor (ICI) abbreviation check it.

 We thank the reviewer for pointing out the need for correction (line 13). 

  1. Line 75, CTLA-4 (cytotoxic T-lymphocyte associated protein 4) the other name called CD152 needs to be change the sentence.

 We thank the reviewer for pointing out the need for correction (line 109-110). 

  1. Line 83 the word “naïve” have the vowel “i” any special meaning?

“Naïve” is the correct spelling.

  1. The missing part of this review is a table. For better understanding, for reader there must be one table. For reference, below review article for references and it will help you to make a table.

10.3390/ijms241310546,

10.1038/s41568-019-0224-7,

10.3389/fimmu.2021.676301

We thank the reviewer for their helpful suggestion of incorporating a table and the additional references for improved understanding.  The original version did include a table (listing clinical trials of small molecule inhibitors in HNSCC) at the end of that document. That table (now Table 2) has now been supplemented with an additional table (Table 1) describing relevant clinical trials using immune checkpoint inhibitors.

  1. If the clinical trial has a better explanation with all details it will be better follow the articles links in comment 4 and try to elaborate. 

We thank the reviewer for their helpful suggestion. Table 1 now describes relevant clinical trials using immune checkpoint inhibitors in HNSCC. Table 2 now describes clinical trials of small molecule inhibitors in HNSCC.

Reviewer 3 Report

This review provide a nice view on the role of neutrophils extracellular traps as new biomarker to evaluate the poor or good response of HNC to immunotherapy.

Also the role of small molecule immunotherapies is well described.

I would add only one more paragraph to render this review more complete:

I suggest to authors to add a paragraph describing the ongoing clinical trials using immunotherapies alone or combination of immunotherapies with other immunotherapies or target therapies for the treatment of HNC in general and not only confined to SMI and or NET. Include also a summary table

In this way the reader can understand the state of art of current clinical trials for HNC and the potential combination of SMI and NET therapy with immunotherapies in general to increase their efficacy for the treatment of HNC

Author Response

  1. This review provide a nice view on the role of neutrophils extracellular traps as new biomarker to evaluate the poor or good response of HNC to immunotherapy. Also the role of small molecule immunotherapies is well described.

We thank the reviewer for their encouraging comments.

  1. I would add only one more paragraph to render this review more complete: I suggest to authors to add a paragraph describing the ongoing clinical trials using immunotherapies alone or combination of immunotherapies with other immunotherapies or target therapies for the treatment of HNC in general and not only confined to SMI and or NET. Include also a summary table. In this way the reader can understand the state of art of current clinical trials for HNC and the potential combination of SMI and NET therapy with immunotherapies in general to increase their efficacy for the treatment of HNC.

We thank the reviewer for their helpful suggestion. The revised manuscript describes recent clinical trials (lines 285-503) and provides a summary table (Table 2) in respect of immunotherapies alone or combination of immunotherapies with other immunotherapies or target therapies for the treatment of HNC in general and not only confined to SMI and or NET. 

Round 2

Reviewer 1 Report

The responses and revisions have significantly strengthened the scientific rigor and clarity of the manuscript. I am confident that the work will be of great interest to the readership and will contribute to advancing the field of head and neck cancer therapeutics.

Reviewer 2 Report

The manuscript is ready for publication.